# HDT Bitmap Triple Indices for Efficient RDF Data Exploration

Maximilian Wenzel[1], Thorsten Liebig[2], and Birte Glimm[1]

[1] Institute of Artificial Intelligence, University of Ulm, Germany,
{maximilian.wenzel, birte.glimm}@uni-ulm.de
[2] derivo GmbH, Ulm, Germany, liebig@derivo.de

**Abstract.** The exploration of large, unknown RDF data sets is difficult even for users who are familiar with Semantic Web technologies as, e.g., the SPARQL query language. The concept of faceted navigation offers a user-friendly exploration method through filters that are chosen such that no empty result sets occur. Computing such filters is resource intensive, especially for large data sets, and may cause considerable delays in the user interaction. One possibility for improving the performance is the generation of indices for partial solutions. In this paper, we propose and evaluate indices in form of the Bitmap Triple (BT) data structure, generated over the Header-Dictionary-Triples (HDT) RDF compression format. We show that the resulting indices can be utilized to efficiently compute the required exploratory operations for data sets with up to 150 million triples. In the experiments, the BT indices exhibit a stable performance and outperform other deployed approaches in four out of five compared operations.

## 1 Introduction

Exploring large data sets is increasingly important in many scenarios. In the context of Semantic Web technologies, however, the exploration of a large, unfamiliar RDF data set can be very challenging, in particular, for novice users of Semantic Web technologies. Faceted navigation is an approach to ease the exploration process by providing users with filters for a given set of resources such that it is guaranteed that applying the filters yields a non-empty result.

Several approaches and systems support faceted navigation: *Ontogator* [9], /facet [8], and *BrowseRDF* [15] are among the first text-based, faceted browsers, where the latter provides metrics for an automatic ranking for the available filters. *GraFa* [13] targets large-scale, heterogeneous RDF graphs and employs a materialisation strategy and a full-text index for queries to improve the response time. The browsers *SemFacet* [2,10] and *Broccoli* [3] support faceted search, i.e., they combine faceted navigation with full-text search. Apart from text-based faceted browsers, there are also graph-based visualisation systems such as *gFacet* [7] and *SemSpect* [11].

The operations needed for faceted navigation are, however, very costly, especially for large data sets. To improve the performance of the required operations, most (if not all) of the above systems use indices. Sometimes, such indices require, however, even more space than the original data set. Therefore, the development of techniques for keeping such indices as small as possible is of great importance. An established

compression format for RDF data is *Header-Dictionary-Triples* (HDT) [12,5], where each resource of the data set is assigned a unique integer ID in a dictionary component. These IDs are then used to generate a compressed representation of all RDF triples in a binary data structure called *Bitmap Triples* (BT). HDT further has the advantage that triple patterns can efficiently be resolved over the BT data structure.

In this work, we propose two BT indices on top of an HDT file for efficiently executing the operations required in faceted navigation. For this, we extend the original BT definition to allow for representing BT indices over subsets of the original RDF graph. An exploration over a given HDT file is especially useful since the HDT format is intended to be an exchange format for RDF data sets. Therefore, using an exchanged HDT file as a basis for a faceted navigation back end might be practical in order to get an overview of an initially unknown data set. In an empirical evaluation, we demonstrate that the generation of these indices is feasible for data sets with up to 150 million triples and that the time and space needed for generating the indices is less than what is required for generating the original HDT file. Based on the indices, a stable and consistently improved performance can be observed compared to executing the operations over the HDT file itself and other in-memory triple stores.

Section 2 introduces the concept of and operations needed for faceted navigation as well as the fundamentals of the HDT format. Section 3 then introduces the novel BT indices and their use in the required exploratory operations. In Section 4, we empirically evaluate our approach and other existing approaches over various real-world data sets. Section 5 concludes the paper.

## 2  Preliminaries

Before going into the details of our proposed indices, we introduce the operations needed for faceted navigation, the basics of Header Dictionary Triples and the Bitmap Triples compression format.

### 2.1  Faceted Navigation

Faceted navigation is an approach to retrieve data which is organised in facets and facet values. In this paper, we consider faceted navigation over RDF graphs:

**Definition 1 (RDF).** *Let $I$, $L$, and $B$ be pairwise disjoint infinite sets of IRIs, literals, and blank nodes, respectively. A tuple $(s, p, o) \in (I \cup B) \times I \times (I \cup L \cup B)$ is called an RDF* triple*, where $s$ is the* subject*, $p$ is the* predicate*, and $o$ is the* object*. An RDF graph $G$ is a finite set of RDF triples. We set $S_G = \{s \mid (s, p, o) \in G\}$, $P_G = \{p \mid (s, p, o) \in G\}$, $O_G = \{o \mid (s, p, o) \in G\}$.*

*Given an RDF graph $G$, we define the set $C_G$ of* classes *as containing all resources $C$ such that $G$ RDFS-entails the triple $(C, rdf{:}type, rdfs{:}Class)$. We say that a resource $s$ has* type *$C$ if $G$ RDFS-entails the triple $(s, rdf{:}type, C)$ and the* instances *of a class $C$ are all resources that have the type $C$ in $G$.*

*In the context of faceted search, for a triple $(s, p, o) \in G$, we also call $p$ a* facet*, and we call $s$ and $o$* facet values*.*

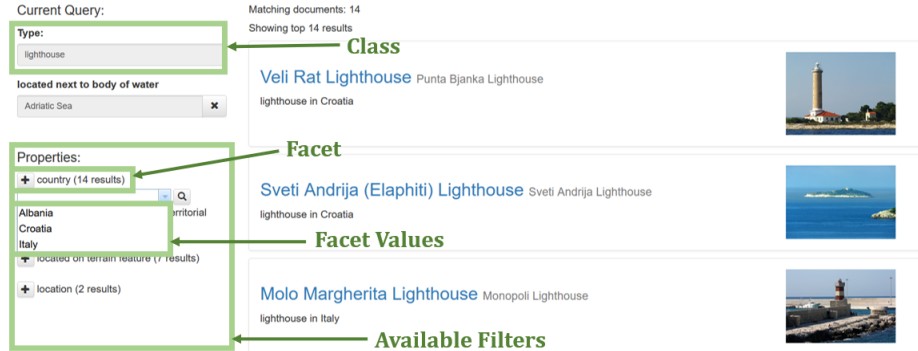

**Fig. 1.** Screenshot of the GraFa [13] application. Proceeding from an initial class, the user refines the set of instances by the step-wise selection of available filters.

We normally omit "RDF" in our terminology if no confusion is likely and we abbreviate IRIs using the prefixes *rdf*, *rdfs*, and *owl* to refer to the RDF, RDFS, and OWL namespaces, respectively.

The initial step in the faceted exploration of a graph is to filter resources according to classes. Once a user has chosen a class, the instances of the class constitute the *centre* of the exploration. This centre is then step-by-step refined by applying filters.

**Definition 2 (Filters).** *Let G be an RDF graph, a* centre $M \subseteq S_G \cup O_G$ *w.r.t. G is a subset of the resources in G.*

*Let V be a countably infinite set of* variables *disjoint from $I \cup L \cup B$. In order to distinguish variables from resources, we prefix variables with a* ?. *A* triple pattern *is a member of the set $(I \cup B \cup V) \times (I \cup V) \times (I \cup L \cup B \cup V)$.*

*An* incoming property-value filter *is a triple pattern of the form $(s, p, ?o)$, while an* outgoing property-value filter *is of the form $(?s, p, o)$. Given a centre M w.r.t. G, applying the property-value filters $(s, p, ?o)$ and $(?s, p, o)$ to M yields $\{o \mid o \in M, (s, p, o) \in G\}$ and $\{s \mid s \in M, (s, p, o) \in G\}$, respectively.*

*A* property filter *is a triple pattern of the form $(?s, p, ?o)$. Applying $(?s, p, ?o)$ to a centre M yields a set of* incoming resources $M_p^{\rightarrow} = \{s \mid o \in M, (s, p, o) \in G\}$ *and a set of* outgoing resources $M_p^{\leftarrow} = \{o \mid s \in M, (s, p, o) \in G\}$.

*Given a centre M w.r.t. G, a filter f is* available, *if applying f over M yields a non-empty result.*

In faceted navigation, only available filters must be presented to the user. The above defined notions and operations are used, for example, in the application GraFa [13] as shown in Figure 1.

Since the exploration is also often based on classes and their instances, which are then filtered, another important concept is that of *reachable classes*:

**Definition 3 (Reachable Classes).** *Let G be an RDF graph and M a centre. A class C induces a set of* incoming (outgoing) facets $\{p \mid (s, p, o) \in G, s \in M, o \text{ has type } C\}$ $(\{p \mid (s, p, o) \in G, o \in M, s \text{ has type } C\})$ *w.r.t. M. We say that C is an* incoming (outgoing) reachable class *w.r.t. M, if the induced set of incoming (outgoing) facets is non-empty.*

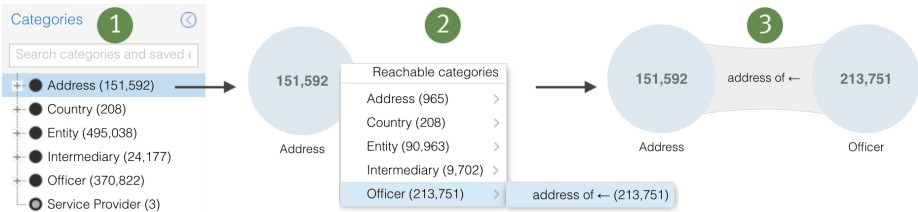

**Fig. 2.** Excerpt from the SemSpect [1] application exploring the Panama Papers data set. (1) A user selects an initial class (in SemSpect denoted as "category"), which then represents the *centre*. (2) Proceeding from the *centre*, all reachable classes are computed. (3) The user chooses one of the reachable classes which subsequently constitutes the new *centre*.

If the direction (incoming/outgoing) is clear from the context, we simply speak about reachable classes.

From a centre $M$, a user may choose a reachable class $C$ and an appropriate facet $p$ to continue the exploration. As a result, an *incremental join operation* is performed to join the incoming (outgoing) resources with the instances of the class $C$. Subsequently, this set becomes the new centre. From there, the available filters and the reachable classes are computed again.

As an example for the reachable classes and incremental join operation, an excerpt from the application SemSpect [1] is shown in Figure 2.

## 2.2 RDF Compression

Traditional RDF serialisation formats (e.g., RDF/XML, Turtle, or N3) have a high level of redundancy and a simple lookup requires a sequential scan of the document. The binary Header Dictionary Triples (HDT) [5,12] format addresses these shortcomings by providing a compressed triple structure that can be queried without the need of decompression. An HDT file is composed of three parts: (i) the *Header* provides metadata about the data set in plain RDF format, (ii) the *Dictionary* provides a mapping between strings and unique IDs, and (iii) the *Triples* encode the RDF graph using IDs. We next consider the concrete implementation of the dictionary and the triples component in more detail as they constitute the basis of the proposed indices.

**Definition 4 (HDT).** *Let $SO_G = S_G \cap O_G$ denote the set of* shared resources*, $S_{pure} = S_G \setminus SO_G$ the set of* pure subjects*, and $O_{pure} = O_G \setminus SO_G$ the set of* pure objects*. For a set $S$, we use $\overline{S}$ to denote the lexicographic order of elements in $S$.*

*The* HDT dictionary *provides a mapping $\mu$ such that $\mu$ maps a resource $r_i$ at position $i$ (starting with position 1) of a list $L$ to $i$ if $L = \overline{SO_G}$, to $|SO_G| + i$ if $L = \overline{S_{pure}}$, to $|SO_G| + i$ if $L = \overline{O_{pure}}$, and to $i$ if $L = \overline{P_G}$.*

Note that a given ID can belong to different sets, but the disambiguation of the correct set is trivial when we know the position (subject, predicate, or object) of the ID in a triple. Furthermore, the distinction into four sets helps to assign shorter IDs since the entries are stored in a binary sequence, where the length of the sequence depends on

the maximal used ID. The ID 0 is reserved as a wildcard character in triple ID queries. The dictionary is eventually stored using *plain front-coding* [4] – a compression format for lexicographically sorted dictionaries based on the fact that consecutive entries are likely to share a common prefix.

The *Triples component* uses the assigned IDs to form a compressed representation of a graph $G$ using the Plain Triples, Compact Triples or Bitmap Triples [5] format.

**Definition 5 (Plain Triples).** *Given an RDF graph G, the* Plain Triples *representation* $\mathsf{PT}(G)$ *of G is obtained by replacing each triple* $(s, p, o) \in G$ *with* $(\mu(s), \mu(p), \mu(o))$, *where $\mu$ is the HDT dictionary mapping.*

The *Compact Triples* representation of $G$ is obtained from $\mathsf{PT}(G)$ based on adjacency lists of predicates and objects for the subjects:

**Definition 6 (Compact Triples).** *For a subject ID s in the plain triples format* $\mathsf{PT}(G)$ *of an RDF graph G,* $\mathsf{pred}(s)$ *denotes the ordered sequence of predicate IDs in* $\{p \mid (s, p, o) \in \mathsf{PT}(G)\}$; *for a subject ID s and a predicate ID p in* $\mathsf{PT}(G)$, $\mathsf{obj}(s, p)$ *denotes the ordered sequence of object IDs in* $\{o \mid (s, p, o) \in \mathsf{PT}(G)\}$.

*Let* $s_1, \ldots, s_n$ *be the ordered sequence of distinct subject IDs in* $\mathsf{PT}(G)$. *The* Compact Triples *encoding* $\mathsf{CT}(G)$ *of G consists of two ID sequences: The* predicate sequence $S_p = \mathsf{pred}(s_1)0 \ldots 0\mathsf{pred}(s_n)$ *and the* objects sequence $S_o = S_o^1 \ldots S_o^n$, *where each partial object sequence* $S_o^i$, $1 \le i \le n$, *is the concatenation* $\mathsf{obj}(s_i, p_1)0 \ldots 0\mathsf{obj}(s_i, p_m)0$ *and* $p_1, \ldots, p_m = \mathsf{pred}(s_i)$.

Note that the unused ID 0 is used to terminate predicate and object sequences. The *Bitmap Triples* representation of $G$ is obtained from $\mathsf{CT}(G)$ by further compressing the lists with the help of additional bitmap sequences:

**Definition 7 (Bitmap Triples).** *Let G be an RDF graph and* $\mathsf{CT}(G)$ *its compact triples representation such that* $S_p = S_p^1 0 \ldots 0 S_p^n$ *and* $S_o = S_o^1 0 \ldots 0 S_o^m$ *are the predicate and object sequence of* $\mathsf{CT}(G)$, *respectively. Let* $\ell_p^i$ *and* $\ell_o^j$ *denote the length of* $S_p^i$ *and* $S_o^j$, $1 \le i \le n$, $1 \le j \le m$, *respectively.*

*The* Bitmap Triples *encoding* $\mathsf{BT}(G)$ *of G consists of two ID sequences* $\hat{S}_p = S_p^1 \ldots S_p^n$ *and* $\hat{S}_o = S_o^1 \ldots S_o^m$, *obtained from* $S_p$ *and* $S_o$ *by dropping the 0s, and two bit sequences* $B_p$ *and* $B_o$ *such that the length of* $B_p$ *and* $B_o$ *is the same as the length of* $\hat{S}_p$ *and* $\hat{S}_o$, *respectively, and the value at position pos in* $B_p$ *($B_o$) is 1 if pos* $= \ell_p^1 + \ldots + \ell_p^i$ *for some i,* $1 \le i \le n$, *(pos* $= \ell_o^1 + \ldots + \ell_o^j$, *for some j,* $1 \le j \le m$*) and it is 0 otherwise.*

Figure 3, from an example introduced by Fernandez et al. [5], illustrates the different formats. While we omit the dictionary component, the Plain Triples show that the IDs $1 \ldots 3$ are subject IDs and that the ID 1 denotes a shared resource as it also occurs in the object position. The further (pure) objects have the IDs $2 \ldots 6$. The IDs $1 \ldots 4$ also represent predicates. The subject 1 is used with the (distinct) predicates 2 and 3, hence, $\mathsf{pred}(1)$ is 23. Analogously, $\mathsf{pred}(2) = 124$ and $\mathsf{pred}(3) = 3$. In $S_p$ these sequences are separated by a 0 and $S_o$ analogously encodes the objects for each subject–predicate pair. In the Bitmap Triples encoding, the sequences $\hat{S}_p$ and $\hat{S}_o$ simply drop any 0 from $S_p$ and $S_o$. The bit sequence $B_p$ tells us with a value of 1, when a sequence in $\hat{S}_p$ ends, i.e., the first sequence 23 (of length 2) ends at (and including) position 2. The following

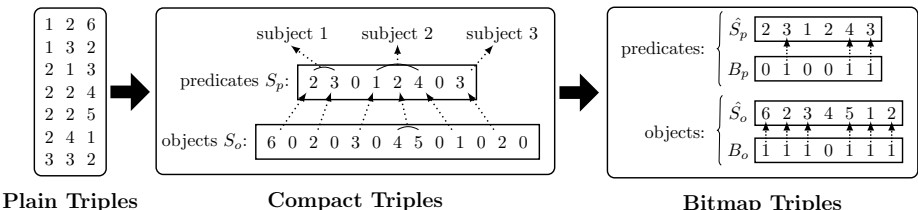

**Fig. 3.** Example for the compressed triple formats from Fernandez et al. [5]

sequence 124 (of length 3) starts at position 3 and the next 1 at position 5 (2 + 3) of $B_p$ indicates the end of the sequence. The bit sequence $B_o$ analogously terminates the sequences in $\hat{S}_o$.

HDT is not merely a compression format but also provides the efficient resolution of triple patterns on the compressed representation of an RDF graph by deploying succinct data structures for the BT implementation [5]. SP-O index queries can be efficiently accomplished in $O(log(n))$, i.e., the triple patterns $(s, p, o)$, $(s, p, 0)$, $(s, 0, o)$ and $(s, 0, 0)$ can be resolved, where 0 denotes a wildcard match and $s, p, o$ represent resource IDs at the corresponding triple position. For other triple patterns, the so far discussed solutions do not suffice and Martínez-Prieto et al. [12], therefore, introduced a compact full-index called *HDT Focused on Querying (FoQ)*, which is created over the original HDT file and which consists of a further PS-O and OP-S index.

## 3   Faceted Navigation Indices Based on HDT BT

In order to efficiently compute the available filters, reachable classes, filter operations and the corresponding incremental joins as described in Section 2, we propose the use of additional (in-memory) BT indices over *subsets* of the original RDF data set. Therefore, in order to represent only a sub-graph, an adaptation of the original Bitmap Triples format (Definition 7) is needed. We illustrate this for the standard SP-O triple component order, but the approach can analogously be transferred to the PO-S and PS-O BT indices. We decided against self-indexing individual triples in the Bitmap Triples component of the HDT file because we need to perform efficient triple pattern queries on the subgraphs in the triple component order PO-S and PS-O. For instance, in case of triple pattern queries $(?s, p, o)$, the self-indexing approach does not require less space since these patterns can only be efficiently resolved by indexing all subjects for the appropriate predicate-object pairs.

**Definition 8  (Partial BT SP-O Index).** *Let G be an RDF graph and $G' \subseteq G$ a subgraph of G. A* partial BT index *for $G'$ consists of* $\mathsf{BT}(G')$ *and the ID sequence $S_s = s_1 \cdots s_n$, where $s_1, \ldots, s_n$ is the ordered sequence of distinct subject IDs in* $\mathsf{PT}(G')$.

For an SP-O index over a sub-graph $G'$ of an RDF graph $G$, the implicit subject ID, which is stored by the bit sequence $B_p$, does not correspond to the actual subject dictionary ID of the original HDT file. Therefore, the additional ID sequence $S_s$ indicates the actual subject ID from the dictionary to each implicit subject ID.

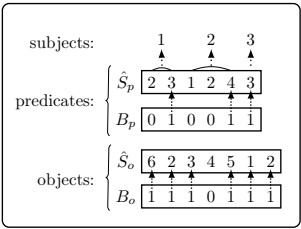 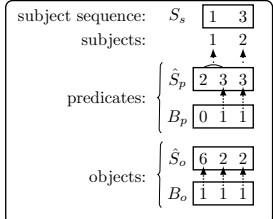

| Bitmap Triples | Partial BT SP-O Index |
| --- | --- |

**Fig. 4.** Bitmap triples from the example in Figure 3 (left-hand side) and for the sub-graph containing triples with subject IDs 1 and 3 (right-hand side) with the additional subject sequence $S_s$ to obtain the original subject IDs

Figure 4 shows again the Bitmap triples from Figure 3 (left-hand side) and the Bitmap Triples for the sub-graph containing only triples with subject IDs 1 and 3. The additional subject sequence $S_s$ is used to obtain the original subject IDs for the HDT mapping.

In order to resolve triple patterns of the form $(s, 0, 0)$ (with 0 as wildcard) on a partial SP-O BT index, the subject ID $s$ is first converted into the implicit subject ID of the corresponding partial BT index. For this purpose a binary search operation is executed on the sequence $S_s$ to obtain the position of $s$ in $S_s$, which corresponds to the implicit subject ID in the bit sequence $B_p$ of $\mathsf{BT}(G')$. Afterwards, the resolution of triple patterns involves the same steps as for the standard Bitmap Triples [12]. However, when iterating over the results from a search over a partial BT index, each subject ID $s_r$ from the search results has to be converted into the appropriate subject ID $s_e$ over the full graph, which can be accomplished by the simple *access* operation $s_e := S_s[s_r]$.

Based on our definition of partial BT indices, we propose the use of two additional (in-memory) BT indices: *Property-Value* and *Class-to-Class* indices.

**Definition 9 (Property-Value Index).** *Let G be an RDF graph with $C \in C_G$ a class. An incoming property value (PV) index w.r.t. C is a partial PS-O BT index over $\{(s, p, o) \in G \mid o$ has type $C\}$. An outgoing PV index w.r.t. C is a PO-S BT index over $\{(s, p, o) \in G \mid s$ has type $C\}$.*

In order to efficiently calculate all available filters and the filter operations, we generate incoming and outgoing PV indices for all classes in an RDF graph. In case of the incoming and outgoing PV indices, the PS-O and the PO-S triple component order has been chosen, respectively, to fetch all relevant triples for a given incoming property-value filter $(s, p, ?o)$ and outgoing property-value filter $(?s, p, o)$, respectively.

In order to efficiently compute reachable classes, we further propose Class-to-Class indices:

**Definition 10 (Class-to-Class Index).** *Let G be an RDF graph with $C, D \in C_G$ classes. A Class-to-Class (CtC) index w.r.t. C and D is a partial PS-O BT index over $\{(s, p, o) \in G \mid s$ has type $C, o$ has type $D\}$.*

**Table 1.** Data sets utilised in the experiments

| Data Set | RDF Dump File (.ttl) | #Triples | #Classes |
|---|---:|---:|---:|
| **LinkedMDB**[5] | 282 MB | 3,579,532 | 41 |
| **Lobbying Filings**[6] | 306 MB | 5,344,200 | 7 |
| **Panama Papers**[7] | 1,140 MB | 19,903,231 | 18 |
| **Reactome**[8] | 1,350 MB | 48,556,891 | 82 |
| **SciGraph**[9] | 8,500 MB | 39,692,376 | 20 |
| **YAGO 2**[10] | 9,920 MB | 158,991,568 | 373,442 |
| **OpenPermID**[11] | 11,470 MB | 152,527,813 | 21 |

We generate CtC indices for each pair of classes $C, D \in C_G$. The PS-O triple component order facilitates the computation of incremental join operations between instances of the classes $C$ and $D$ since all existing triples in the corresponding index can efficiently be retrieved for a given facet $p$.

## 4 Implementation and Evaluation

We next present the empirical evaluation of the proposed additional BT indices. We start by describing used data sets as well as the time and space required to create and store the proposed BT indices for these data sets. We then introduce the different benchmarked approaches and the queries and tasks used in the evaluation before we present the actual results.

All presented experiments were conducted on an Intel(R) Core(TM) i7-3930K CPU @ 3.20GHz, 6 cores, 64GB DDR3 @ 1334MHz.

### 4.1 Data Sets and Index Generation

The (partial) BT indices were implemented in Java with the HDT Java library[3] as foundation and the code is available on GitHub as open-source project.[4] Since our implementation does not support RDFS-entailment, the required inferences (for types of the resources) are materialised upfront using RDFox [14]. For each RDF file, we generate an HDT file, which serves as the basis for the CtC and PV indices. Note that RDF classes, which do not contribute to the semantic domain of the RDF model such as *rdfs:Resource*, are not considered in the indices.

In order to get an impression of the required generation time and resulting file size, the proposed BT indices have been generated for various data sets (see Table 1). For the YAGO 2 data set, the CtC and PV indices could not be generated because of insufficient main memory capacity for the high number of distinct RDF classes.

---

[3] `https://github.com/rdfhdt/hdt-java`

[4] `https://github.com/MaximilianWenzel/hdt-bt-indices-java-lib`

[5] `http://www.cs.toronto.edu/~oktie/linkedmdb/linkedmdb-18-05-2009-dump.nt`

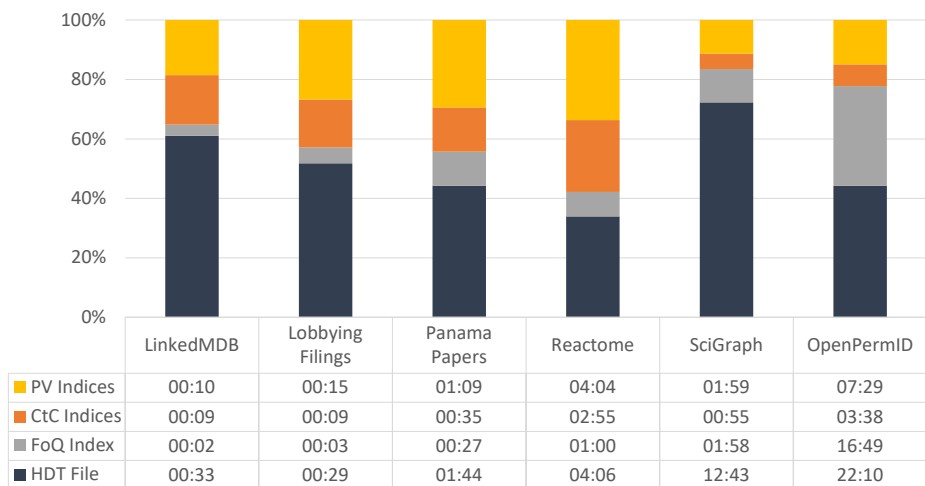

| | LinkedMDB | Lobbying Filings | Panama Papers | Reactome | SciGraph | OpenPermID |
|---|---|---|---|---|---|---|
| PV Indices | 00:10 | 00:15 | 01:09 | 04:04 | 01:59 | 07:29 |
| CtC Indices | 00:09 | 00:09 | 00:35 | 02:55 | 00:55 | 03:38 |
| FoQ Index | 00:02 | 00:03 | 00:27 | 01:00 | 01:58 | 16:49 |
| HDT File | 00:33 | 00:29 | 01:44 | 04:06 | 12:43 | 22:10 |

**Fig. 5.** Time required for generating the representations (format mm:ss)

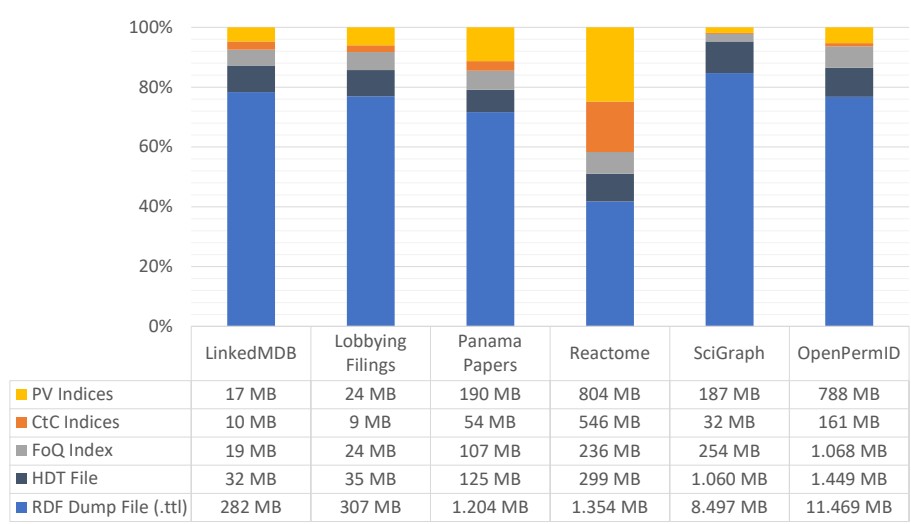

| | LinkedMDB | Lobbying Filings | Panama Papers | Reactome | SciGraph | OpenPermID |
|---|---|---|---|---|---|---|
| PV Indices | 17 MB | 24 MB | 190 MB | 804 MB | 187 MB | 788 MB |
| CtC Indices | 10 MB | 9 MB | 54 MB | 546 MB | 32 MB | 161 MB |
| FoQ Index | 19 MB | 24 MB | 107 MB | 236 MB | 254 MB | 1.068 MB |
| HDT File | 32 MB | 35 MB | 125 MB | 299 MB | 1.060 MB | 1.449 MB |
| RDF Dump File (.ttl) | 282 MB | 307 MB | 1.204 MB | 1.354 MB | 8.497 MB | 11.469 MB |

**Fig. 6.** File size comparison of the various representations

---

[6] We are not authorized to publish the data set in RDF format. REST API for XML download: `https://lda.senate.gov/api/`, accessed: 2019-10-15

[7] `https://doi.org/10.5281/zenodo.4319930`

[8] `https://doi.org/10.5281/zenodo.4415888`

[9] `https://www.springernature.com/gp/researchers/scigraph`

[10] `http://yago.r2.enst.fr/`

[11] `https://permid.org/`

In Figure 5, the time required to generate all CtC and PV indices is compared to the time needed to create the HDT file and the compact HDT Focused on Querying (FoQ) full-index [12]. Figure 6 shows the size of the BT indices compared to the size of the original RDF dump file, the HDT file and the HDT FoQ index. Neither the time for generating the CtC nor the PV index exceeds the time for generating the HDT file on any data set. The files that are utilised in an actual faceted exploration, i.e., the PV, CtC, and FoQ indices and the HDT file, require between $18 - 30\%$ of the Turtle file size in five out of six data sets. Only for the Reactome data set, 139% of the Turtle file size is needed which is due to the high interconnectivity, i.e., the high number of triples $t = (s, p, o)$ where $s$ and $o$ have an RDF class assigned. The consumed storage space subsequently corresponds to the required main memory capacity during an exploration.

### 4.2 Performance Benchmark Setup

We consider the following approaches in the experimental evaluation:

1. *HDT Jena:* The Jena ARQ query engine[12] used over an HDT file.
2. *RDFox:* The commercial, in-memory triple store RDFox [14].
3. *Plain HDT:* Our implementation using only the original HDT file and the FoQ index.
4. *PV indices:* Our implementation used with PV indices, the HDT file and the FoQ index. The FoQ index is utilized to perform efficient triple pattern queries that enhance the property-value filter and available filter operations if a given centre $M$ has less resources than a threshold $\theta$, which was experimentally determined [16]. For instance, in case of property-value filter operations, $\theta = 5,000$.
5. *CtC indices:* Our implementation used with CtC indices and the HDT file.
6. *Hybrid:* Our implementation used with PV indices, the CtC indices, the HDT file and the FoQ index.

Our evaluation initially also included other triple stores. However, we limit our result presentation to a comparison with RDFox and HDT Jena as these were the best performing systems for the kinds of queries needed in faceted exploration. When interpreting the results, it should be noted that RDFox and HDT Jena offer full SPARQL 1.1 support and RDFox further supports incremental reasoning. This is in contrast to our HDT BT indices which are particularly optimized but also limited to faceted query answering. Our approach also comes with a single upfront HDT BT indices creation time. For instance, the largest considered data set, OpenPermID, can be loaded query ready in 93 $s$ into an RDFox data store, whereas the generation of our CtC and PV indices alone requires about 11 $m$.

The used queries and tasks in the experiments are designed to fulfill the following two requirements: (i) The queries should be representative for the data set. We use the principle of stratified randomisation to design queries, which correspond to a specific level of difficulty. Since it was not always possible to generate sufficient queries for all difficulty levels, the actual number of queries in an experiment may vary depending on the data set. We comment on this in the evaluation results. (ii) The queries should be

---

[12] https://jena.apache.org/documentation/query/index.html

representative for a faceted navigation scenario. To address this, we use the following query types and tasks in our benchmark:

1. *Filter queries* apply a sequence of property-value filters to a given centre $M$. Since incoming and outgoing filters require the same computational effort using the PV indices, only outgoing filters are considered. As result, all IRIs of the resources from the filtered centre are returned. Overall, we consider 1 to 3 filters over 3 levels of difficulty with 10 queries per difficulty level, where the relevant parameters for the difficulty level comprise the size of $M$ and size of the result set obtained by applying the filter. Thus, the maximal number of evaluated filter queries per data set is $n_{max} = 3 \cdot 3 \cdot 10 = 90$.

2. The task of *computing all available filters* is based on a filter query, which is used to initialise a centre $M$. Relative to this centre, all available incoming and outgoing property-value filters are computed. As result, the number of distinct facet values for each facet is subsequently returned. Overall, we use 1 filter over 3 levels of difficulty with 10 queries per difficulty level, i.e., we evaluate at most $n_{max} = 30$ task executions per data set.

3. *Class queries* start with a given initial centre $M_0$ and perform $n$ incremental join operations to an incoming or outgoing reachable class. All IRIs of the resources from the final centre $M_n$ are subsequently returned. The number of joins $n$ corresponds to the number of used facets. The parameters, which were considered in the query generation process for the corresponding level of difficulty, are again the size of $M_0$ and, on the other hand, the number of triples that participate in the join operations. We evaluate class queries that use 1 to 4 joins over 3 levels of difficulty with 10 queries per difficulty level, i.e., we evaluate at most $n_{max} = 120$ task executions per data set. The incremental join operations are explicitly indicated in the resulting SPARQL queries by using nested subqueries with single result variables to lower the number of intermediate results and eventually increase the efficiency, i.e., each SPARQL query begins with the resulting centre $M_n$ and the preceding class $M_{n-1}$ is defined in an appropriate subquery which in turn uses a subquery for its respective predecessor $M_{n-2}$ if present.

4. The task of *computing all reachable classes* starts from a centre of resources, obtained using a class query, and computes the reachable classes relative to this centre. The returned result is the number of reachable classes, i.e., the number of incoming and outgoing facets w.r.t a reachable class as described in Definition 3. We use class queries with 1 join over 3 levels of difficulty and 10 queries per difficulty level. Overall, we evaluate at most $n_{max} = 30$ task executions per data set.

5. *Hybrid queries* combine class and filter queries. We obtain a centre $M$ by executing a class query (which requires a series of joins) and then apply a filter query to the obtained centre. All IRIs of the resulting resources are returned. Again, we consider 3 levels of difficulty which coincide with the number of applied filters and the difficulty level of the underlying class query. We evaluate hybrid queries that use 1 to 4 joins over 3 levels of difficulty with 10 queries per difficulty level, i.e., we evaluate at most $n_{max} = 120$ task executions per data set.

Note that all operations in our experiments compute exact results which is of course more expensive than the calculation of partial results. Partial results, however, often

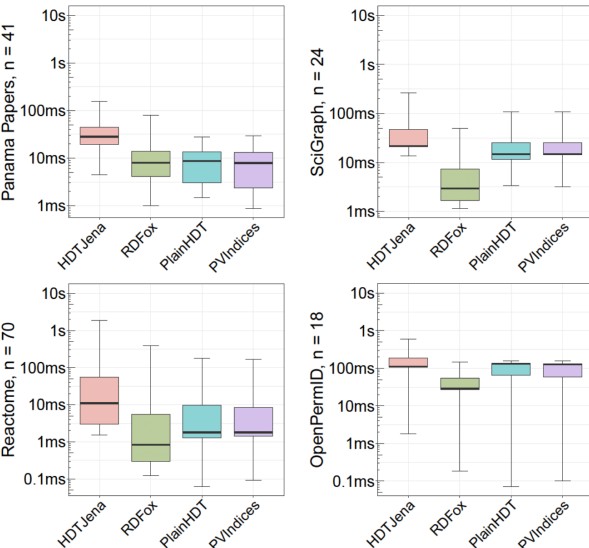

**Fig. 7.** Results of the filter query experiment

increase the scalability of the eventual application, which is observed, for instance, in the SPARQL query builder *SPARKLIS* [6].

The complete execution of an experiment comprises the following steps: First, the query engine is initialised and the queries are loaded from a file into main memory. Second, a warm-up round begins where all queries are executed before the actual experiment. Afterwards, the execution time of all queries is measured in two consecutive rounds and the minimum required time is documented. If a query requires more than ten seconds for the execution, a timeout occurs and is documented correspondingly. A timeout of ten seconds has been chosen because an execution time beyond ten seconds is unreasonable concerning the user-experience in an exploratory setting.

### 4.3 Experimental Results

In the following sections all results from the experiments are presented. The collected time measurements were evaluated using the programming language R and are summarized by box plots. The value $n$ represents the actual number of queries from each experiment. Recall that not for all difficulty levels sufficiently many queries could be generated and, hence, $n \leq n_{max}$. We present the experimental results of the four largest data sets from Table 1 for which the PV and CtC indices could be generated. The full experimental results and queries are available online [16].

Figure 7 shows the results of the *filter queries*. The PV indices and the Plain HDT approach perform equally well on all data sets – no considerable benefits are obtained using additional PV indices on top of the original HDT file. Apart from a few outliers, e.g., on the Reactome data set, RDFox is the leading approach for the resolution of filter queries with reference to the median since it is the lowest observed on all data

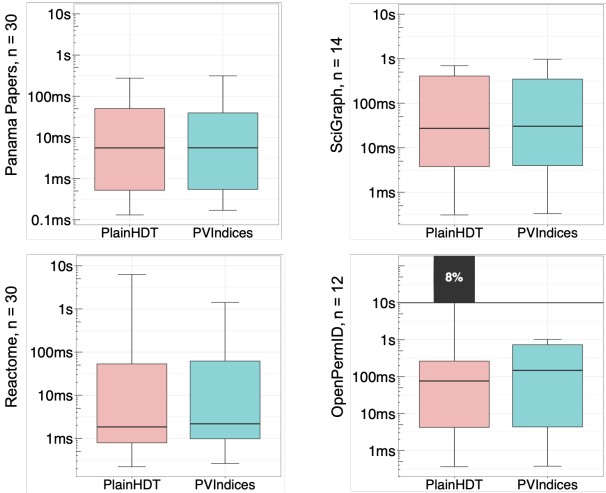

**Fig. 8.** Results of the available filters experiment

sets. RDFox and the Jena ARQ query engine both utilise query planning in contrast to Plain HDT and the PV indices. The performance of our implementation could certainly be improved with additional query planning, which is a topic for future research.

Figure 8 shows the results of the *available filters* experiment. Since the available filters are computed relative to the current *centre*, we only compare the PV indices approach with the Plain HDT approach. A fair comparison with the other approaches, which execute SPARQL queries, would be difficult in this scenario. Apparently, the Plain HDT approach and the PV indices perform generally well in that 50% of the queries always require less than 1 *s*. For larger data sets such as Reactome and OpenPermID, the Plain HDT approach, however, reveals significant performance issues, where occasionally a query requires more than 5 *s* in case of the Reactome data set and in case of the OpenPermID data set even a timeout occurred. In contrast, the PV indices approach constantly requires less than 2 *s* and shows stable performance across all queries.

The results of the *class queries* are shown in Figure 9. Concerning the HDT Jena approach, at least 21% timeouts occurred on all data sets. Although RDFox requires in all cases less than 1 *s* for at least 75% of the queries, several outliers appear in case of the SciGraph and OpenPermID data set, where furthermore 1% of the queries timed out. Plain HDT shows a stable performance in case of the Panama Papers data set, but in other cases, such as OpenPermID, 18% timeouts occur and, in sum, about 25% of the queries require more than 5 *s* for the completion. The CtC indices show an overall stable performance with a maximum required execution time on the SciGraph data set at around 3 *s*.

In Figure 10, the results from the *reachable classes* task are presented. Since the reachable classes are computed relative to the current *centre*, we only compare the CtC indices approach with the Plain HDT approach. A fair comparison with the other approaches would be difficult in this scenario. As can be seen, the CtC indices outperform

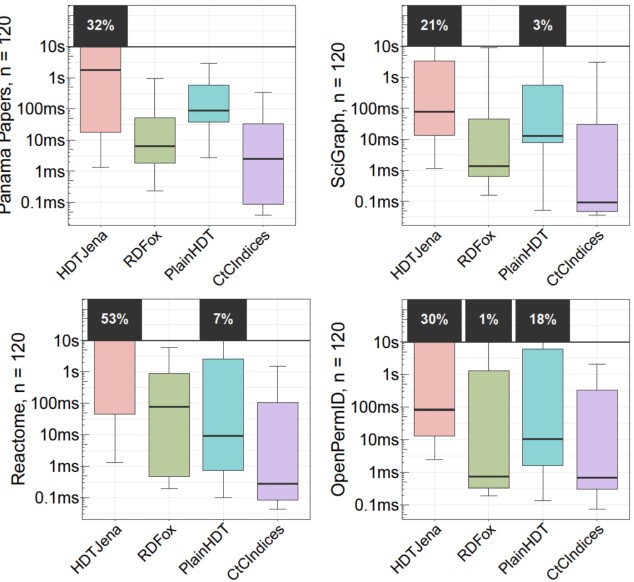

**Fig. 9.** Results of the class query experiment

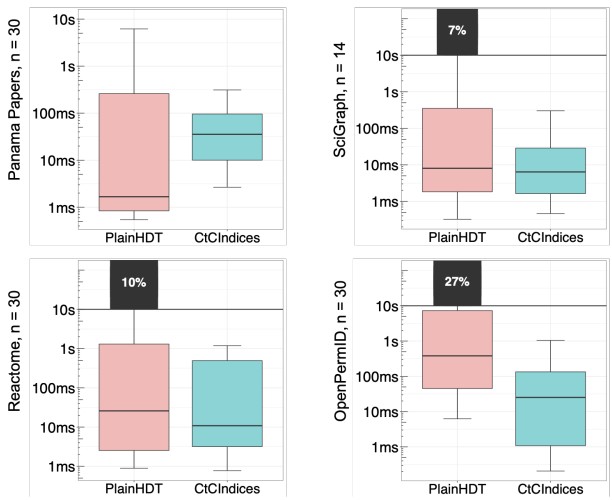

**Fig. 10.** Results of the reachable classes experiment

the Plain HDT approach for all data sets. Especially concerning the data sets SciGraph, Reactome and OpenPermID, Plain HDT requires in some cases even more than $10\,s$ whereas the CtC indices need at most about $1\,s$ for the execution.

The results of the *hybrid queries* are presented in Figure 11. Those approaches, which utilise query planing, i.e., the Jena ARQ query engine and RDFox, have evident

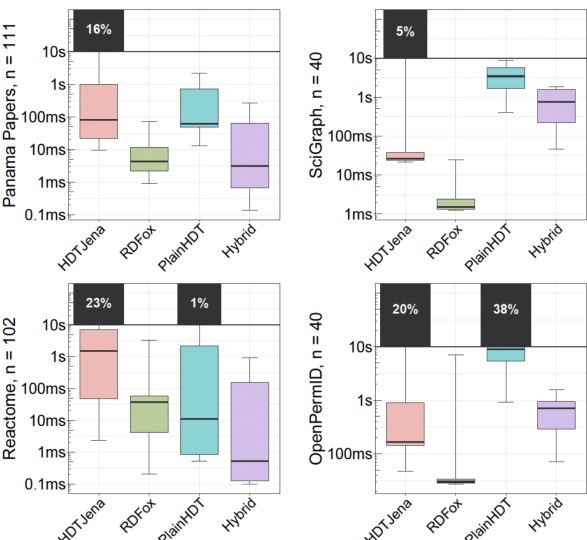

**Fig. 11.** Results of the hybrid query experiment

benefits concerning the resolution of hybrid queries. For instance, in case of RDFox, 75% of the measured times from SciGraph and OpenPermID are considerably lower than those of the Plain HDT and the Hybrid approach. Nevertheless, at least 5% of the queries timed out in case of HDT Jena on all data sets and, in case of Reactome and OpenPermID, RDFox shows a worst case evaluation time of about 3 *s* and 7 *s*, respectively. The maximum execution time of the Hybrid approach is considerably lower on all data sets than the maximum time required by Plain HDT. The Hybrid approach has furthermore a worst case evaluation time of 2 *s*, as can be observed for the SciGraph data set, whereas, e.g., 38% of the queries executed by Plain HDT on the OpenPermID data set led to a time out.

## 5   Conclusion

In this paper, we propose an approach to generate additional indices on top of the RDF compression format HDT for the efficient exploration of RDF data sets. To achieve this objective, the Bitmap Triples (BT) data structure of the HDT file was extended to a BT index which is able to store subsets of the original RDF graph. In order to cover all required exploratory operations of a faceted navigation scenario, two kinds of BT indices have been introduced, namely the Property-Value (PV) and the Class-to-Class (CtC) indices.

Our evaluation over real-world data sets shows that the generation of the CtC and PV indices is feasible for data sets with up to 150 million triples and 82 RDF classes. Neither the generation time of the PV nor of the CtC indices exceeded the duration for generating the original HDT file. Likewise, the file sizes of the PV and CtC indices do not surpass the original HDT file size.

In the performance benchmark, the CtC indices, the PV indices and the Hybrid approach, i.e., the combination of both kinds of indices, show an overall stable performance for all considered exploratory operations across all data sets with a maximum execution time of 3 *s*. We conclude that such BT indices represent a significant contribution for the Semantic Web and faceted navigation in particular and that further improvements are possible, when the approach is combined with other optimisations such as query planning and a memory efficient BT indices generation method.

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
