# OpenReview forum: "HDT Bitmap Triple Indices for Efficient RDF Data Exploration"
_eswc-conferences.org/ESWC/2021/Conference/Research_Track — ESWC 2021 Research_

### Official Review · AnonReviewer3 · 2021-01-13
**The paper presents an interesting theoretical and practical approach towards building a fast RDF faceted system.**

**Rating:** 3
**Confidence:** 5
**Impact:** 5
**Design And Technical Quality:** 5

**Review:**

The paper is concerned with the rapid exploration of RDF datasets. The lack of fast indexing and navigation methods has been the main reason for the slow adoption. Another reason is the complexity of SPARQL which is still an issue for many database programmers.

The paper is rather well-written and illustrated. The experimental section is also interesting and the results are impressive.

The paper is missing a separate Related Work section, but some pointers to the competing tools are provided in the introduction. Instead it offers a rather long Preliminaries chapter. While I do understand why the authors have written the paper as they have and generally like the end result, the rather long preliminary section might end up being uninteresting for readers interesting in the practical aspects of the paper (e.g., evaluation). A more compressed approach for Section 2.1 - e.g., a table presenting the concepts as it is typically done in ML - could have delivered the same information without so many long definitions. However I would not recommend changing the paper at this point, as the definitions were rather easy to follow in the end.

Section 2.2. on RDF Compression also reads like a primer and is really good. I would have added more images here, but I think it is okay also if the readers make an effort and read one or two other papers to fully grasp these concepts.

The faceted navigation indices section is rather well-written and quite compact. This proves that indeed the long introduction was necessary.  Again, I would have loved one or two additional illustrations here - but it is fine also in its current form :).

Dataset and index Evaluation section is probably the most impressive of the paper as it clearly offers all the information needed for reproduction (e.g., even number of triples/classes from each dataset or sizes for the various indices). Also a detailed explanation is offered for each of the tasks.

The results are impressive - but this can still only be used mostly as a faceted navigation solution. To actually build a search engine on top of this solution would still be daunting - as 1-3 seconds (the maximum execution time claimed for the hybrid solution) is a lot of time for getting a good answer, whereas most search engines would need subsecond query times (including the time to build the visual interface or compute/reuse various aggregation results).







**Anonymity:**

No, I would like my review to be deanonymized.

**Reuse And Availability:**

5: Very High

**Strong Points:**

- the paper is well-written and mostly well-illustrated
- the paper presents both the theoretical approach and the practical results obtained
- the evaluation is stellar

**Subreviewer:**

I submitted this review.

**Weak Points:**

- the Preliminaries section could have been condensed a little bit (e.g., through tables, figures, etc)
- there are less details on competing systems

---

> ### Author Rebuttal · Authors · 2021-01-29
>
>
>     "The paper is missing a separate Related Work section, but some pointers to the competing tools are provided in the introduction. Instead it offers a rather long Preliminaries chapter."
>
> Yes, we shortened the related work section in favor of providing a mostly self-contained paper. We will make an effort to condense the preliminaries a bit to make room for a dedicated related work section.
>
>     "The faceted navigation indices section is rather well-written and quite compact. This proves that indeed the long introduction was necessary. Again, I would have loved one or two additional illustrations here - but it is fine also in its current form :)."
>
> Thank you! We would have included more illustrations but it was difficult to stick to the limitation of 15 pages...
>
>     "The results are impressive - but this can still only be used mostly as a faceted navigation solution. To actually build a search engine on top of this solution would still be daunting - as 1-3 seconds (the maximum execution time claimed for the hybrid solution) is a lot of time for getting a good answer, whereas most search engines would need subsecond query times (including the time to build the visual interface or compute/reuse various aggregation results)."
>
> Yes, indeed, 1-3 seconds still is some time. It also depends what kind of results are computed: result estimations, partial results or exact results. In our case, we always calculate the exact results which, however, requires higher effort. This decision should always be made with reference to the appropriate use case.

---

> > ### Comment · AnonReviewer3 · 2021-02-05
> > **rebuttal**
> >
> > I have read the rebuttal. My opinion on the paper does not change.
> > I think the authors have done a good job of answering my requests.
> > Lowering the score would not change anything anyway in the situation of this particular paper.
> > To me there isn't a reason to necessarily reject the paper on the grounds that the the system is not the best or that it has a lot of space for improvements. Indeed however, it can be either shifted to a workshop (if there's still that possibility) or moved to a another conference or journal.
> >
> > My preference would be to shift this publication to something like "Findings of the ISWC" if ISWC/ESWC/EKAW were to decide to organize such sister-publications (see this link from ACL's EMNLP conferences to understand what this kind of publication entails - https://2020.emnlp.org/blog/2020-04-19-findings-of-emnlp).

---

### Official Review · AnonReviewer4 · 2021-01-13
**An HDT-based faceted navigator**

**Confidence:** 5
**Impact:** 4
**Design And Technical Quality:** 3

**Review:**

This paper proposes a faceted navigator built on top of enhanced HDT-based files. In practice, the current approach precalculates solutions of navigation filters and indexes them using an enhanced variant of HDT bitmaps that allows the corresponding subgraphs to be encoded with respect to the HDT dictionary. The experiments reports good (but not impressive) numbers in performance, although the analysis of space requirements are not clear for me.

The paper is easy to read, motivates the relevance of faceted navigation and provides a set of preliminary definitions that make it self-contained and easy to understand for non-experts. Besides, all source code is publicly available, making the paper reproducible.

I have an important doubt after reading the paper: does the current approach need the original Bitmap Triples to provide faceted navigation or only performs on Property-Value and Class-to-Class indexes? I think it is the latter because search operations seem to be performed only on these structures (and the HDT Dictionary), but clarify it please.

The approach is carefully explained in Section 3 and the proposed Property-Value and Class-to-Class indexes are defined. Both structures are built on top the so-called Partial BT SP-O index, that is an enhanced HDT Bitmap structure with an additional integer sequence. These structures index the subgraphs that match the corresponding filters, so they index a subset of triples from the original graph. Thus, the additional integer sequence is used to encode explicitly subject-predicate-object trees present in the subgraph.

-	Although effective, I think that more competitive tradeoffs could be achieved if you replace the current ID sequence by a bitmap B that sets bits corresponding to subjects in PT(G’). This way, you can map the original subject value (i) to the local one by performing $rank_1(B, i)$, which counts the number of 1s until the $i$-th position, and decoding the original ID from a local one (j) by simply performing $select_1(B, j)$, which returns the position of the $j$-th 1 in B.
-	You can even obtain additional tradeoffs if you encode Property-Value and Class-to-Class indexes as bitmaps on top of the original HDT Triples; i.e. one bitmap of length $n$ (where $n$ is the number of total triples) for each index.  Note that leaves in HDT Triples encodes each triple in the graph, so you can easily indexes a subgraph setting the bits corresponding to the triples that match property-value and class-to-class conditions ($i$-th triple in HDT is encoded by the path that ends in the $i$-th leaf). I hope that this will result in sparse bitmaps, which can be compressed using different techniques, depending on the bit distribution. This class of bitmap indexes has been used, for instance, to encode versioned graphs on a compressed representation, reporting good space-time tradeoffs [1].

Although I understand the approach, I miss the algorithms that describe the navigation operations in these indexes. Please, consider including them to improve the paper.

The performed evaluation is interesting, although it must be improved if the paper is finally accepted. Some considerations:

-	Compressed RDF excels for Big Semantic Data but the largest datasets in the current setup is roughly 150 million triples. I suggest to evaluate this approach on larger datasets to analyze possible scalability issues. On the other hand, I think that you should include datasets with a medium number of classes, because I can understand that your approach crashes with YAGO (373K classes) but the other datasets have less than 100 classes.
-	Although interesting, time required for generating the representations is not relevant in this WORM (write once read many) scenario. Anyway, I think that the comparison is unfair because your approach needs HDT files to be previously compressed, so this time should be added to that required to build  your indexes. You should compare your numbers to that reported in the FoQ indexing process of HDT for fairer analysis.
-	I don’t understand the percentual scale in Figure 6. What does 100% mean? Because these bars add the dump file (TTL) size, the HDT file size, the FoQ indexes size and your indexes sizes. Why? You should first clarify the storage requirements of your solution and then compare these numbers with the FoQ ones. This seems fairer to me, because both approaches allow HDT files to be navigated.
-	Can you provide any additional detail about System X? I understand that it is pending of permission, but it is difficult to analyze the impact of your results without knowing its underlying indexes.
-	Choosing Plain HDT in this benchmark is clearly unfair because it only performs efficient subject-based operations, and faceted navigation filters also access by predicate and by object. I can’t accept this evaluation as is, and you must replace Plain HDT by HDT-FoQ to compare to an indexed HDT approach.
-	What do you think about the performance of HDT Jena? It is strange that is performs worse than Plain HDT if it is using a query optimizer. Do you have any insight about it?
-	I suggest including any space-time figure to understand space-time tradeoffs of all approaches. I like to see the difference on space requirements of your approach and HDT-FoQ, and also its query performance, that I expect to improve by large Plain HDT.


Finally, it is worth noting that data structures and algorithms proposed in this paper are publicly available at a GitHub repository, making the paper reproducible.


[1] Ana Cerdeira-Pena, Antonio Fariña, Javier D. Fernández, Miguel A. Martínez-Prieto: Self-Indexing RDF Archives. DCC 2016: 526-535


**Anonymity:**

No, I would like my review to be deanonymized.

**Rating:**

-1: Weak Reject

**Reuse And Availability:**

4: High

**Strong Points:**

-	The paper is well-written, and it is easy to read. I think this class of papers could be very pedagogical due to shows the benefits of managing and querying compressed RDF in practical terms.
-	The approach is simple but efficient. It leads the comparison in some experiments and shows a stable performance in all of them.
-	Faceted navigators like it could be very useful for the community.


**Subreviewer:**

I submitted this review.

**Weak Points:**

-	Although efficient, the proposed tool does not dominate the comparison in general, and it does not outperform the baseline of Plain HDT in some experiments. Besides, I have doubts about the comparison to HDT, because the paper does not make clear if FoQ indices are used.
-	Experimentation lacks of important figures to understand space-time tradeoffs of the current approach and its comparison to the other tools.
-	Faceted search operations are only explained in words, but the corresponding algorithms are not formalized.

---

> ### Author Rebuttal · Authors · 2021-01-29
>
>      "(...) [D]oes the current approach need the original Bitmap Triples to provide faceted navigation or only performs on Property-Value and Class-to-Class indexes? (...)"
>
> That is a good point and should be definitely included. In the class queries, we do not require the original HDT BT indices or the FoQ index - we only use the CtC indices. On the other hand, in case of the filter queries, we used a heuristic threshold $t$: Given a set of resources $M$. If $size(M) < t$, we use the HDT FoQ index to test the filter condition by executing a triple pattern query for each resource $s \in M$. Otherwise, we iterate over the respective triples from the PV indices and check the condition.
>
>      "Although effective, I think that more competitive tradeoffs could be achieved if you replace the current ID sequence by a bitmap B that sets bits corresponding to subjects in PT(G’). (...)"
>
> This is really an interesting idea. If we index triples in the order PO-S or PS-O, this actually might work very well, since an RDF data set often only has a small amount of predicates - thus, the bitsequence is very short. But in case of an SP-O (or any other order with subject in front), the bitsequence could become very large although the index contains a small amount of distinct subjects.
>
>      "You can even obtain additional tradeoffs if you encode Property-Value and Class-to-Class indexes as bitmaps on top of the original HDT Triples; (...). "
>
> Another really good suggestion. Thank you for the reference. But unfortunately, we cannot resolve triple pattern queries for a given predicate on these subgraphs, i.e., a triple pattern query $(?s, p, ?o)$ cannot be resolved. Either each triple in our index has to be examined if it is connected to predicate p or we must create separate indices for each predicate.
>
>      "Although I understand the approach, I miss the algorithms that describe the navigation operations in these indexes. (...)"
>
> Due to the limitation of 15 pages and the extensive evaluation, we decided to prune the algorithms from the paper. Aside from the conversion of the actual ID to the implicit ID of a given BT index, the steps are the same as for the original HDT bitmap triples.
>
>      "Compressed RDF excels for Big Semantic Data but the largest datasets in the current setup is roughly 150 million triples. I suggest to evaluate this approach on larger datasets to analyze possible scalability issues. (...)"
>
> While compression is indeed a very good technique also for very large datasets, faceted exploration (in particular with exact results as we aim for) is more targeted towards small to medium sized datasets. Nevertheless, executing faceted search operations directly over an HDT file seems useful and we show that with suitable indices also feasible.
>
>      "(...) I think that the comparison is unfair because your approach needs HDT files to be previously compressed, so this time should be added to that required to build your indexes. You should compare your numbers to that reported in the FoQ indexing process of HDT for fairer analysis."
>
> You are right, our approach depends on the HDT file and the FoQ index. We did not intend to compete against their generation time. We merely attempted to give the reader an impression of how much time is roughly required to generate the BT indices for a given HDT file.
>
>      "I don’t understand the percentual scale in Figure 6. What does 100\% mean? (...)"
>
> Figure 6 compares the relative size of the formats - it was included for a rough overview of the required size. The actual size is presented in the table. But your suggestion might also be a possible solution.
>
>      "Can you provide any additional detail about System X? (...)"
>
> We are now authorized to deanonymize the results of System X. The deployed system is RDFox. To their request, we had to change the query format to a semantically equivalent one based on sub-queries. This query formats avoids some timeouts for RDFox, but in most cases, the BT indices still perform better. In the updated version we would include the new results.
>
>      "(...) I can’t accept this evaluation as is, and you must replace Plain HDT by HDT-FoQ to compare to an indexed HDT approach."
> Sorry, it was not explcitly mentioned: Plain HDT uses of course the HDT file and the HDT FoQ index.
>
>      "What do you think about the performance of HDT Jena? It is strange that is performs worse than Plain HDT if it is using a query optimizer. Do you have any insight about it?"
>
> As can be seen in the hybrid and filter experiment, HDT Jena has in some cases significant benefits over our approach because of its query optimizer. But as soon as the number of (necessary) intermediate results increases, it does not perform very well. It is, unfortunately, always difficult to explain another systems result in full detail. We nevertheless consider it useful to include the results and we can of course also be reproduced with the provided data by interested readers.

---

> > ### Comment · AnonReviewer4 · 2021-02-02
> > **Rebuttal acknowledgement**
> >
> > Dear authors, thanks for your clarifications. Then, if I understand it correctly your approach requires additional space on top of HDT-FoQ to perform all queries, is it? I think that it must be explained in the paper if it is finally accepted.

---

### Official Review · AnonReviewer1 · 2021-01-14
**Elegant approach, but needs further work to eventually create a nice paper!**

**Rating:** 1
**Confidence:** 3
**Impact:** 3
**Design And Technical Quality:** 3

**Review:**

The paper is really informative on faceted navigation and on RDF compression techniques.

The authors propose an elegant extension to those data structures for efficient faceted search,

However, those extensions seem trivial as a contribution of the paper, since they only build on existing techniques. In fact, the true contribution of the authors is 1.5 page of text.

Commentary for Fig 5 and Fig 6 could be improved. For example, why 139% of the Reactome turtle file is needed? This in not appropriately explained!

The actual related work is covered only in a single paragraph, without appropriately explaining why this approach is better.

I would expect that the proposed indexes make thing more efficient than previous approaches. However, even this is not appropriately evaluated as in the evaluation the proposed system is not compared with another one.

In the result figures there is some space for additional plots to be added. I tried to download the results from zenodo but they are too many for making any conclusion without help.

Also, I do not see actually the benefit of your approach when compared to approaches without the proposed indexes. In some cases what you propose is better but in others your proposal is worst.

Assuming that you selected to present the best score the experiments show that your proposal need to be certainly improved!

In addition, better explanations are required on why we see those results. Currently the results are presented leaving the reader not knowing why s/he sees the specific results on the plots.

Showing only that your proposal is feasible in most of the cases (you even fail to deliver results for YAGO) is not enough, especially when you don’t have sufficient explanations on why this is happening.

I would try to identify where your approach is better, highlighting this aspect right from the introduction. In addition, your paper could be improved by adding a comparison by another faceted search system somehow.

>Comments after rebuttal:
>I would like to thank the authors for their effort to answer my comments. Although it is now a bit clearer, still I believe the paper could benefit from additional work and experiments and an improvement in the related work. Nevertheless, I'm willing to accept this for publication, given that the authors in the CR version of the paper try to address all reviewers concerns.


**Anonymity:**

Yes, I would like my review to remain anonymous.

**Reuse And Availability:**

4: High

**Strong Points:**

-Interesting topic, elegant solution

-Interesting benchmark for evaluation

**Subreviewer:**

I submitted this review.

**Weak Points:**

- Related work could be improved

-Explanations on the results is not adequate

-Performance can be improved, or at least better highlight what are the benefit over compared approaches.

-A comparison with another faceted search system would enhance the quality of the paper

---

> ### Author Rebuttal · Authors · 2021-01-29
>
>     "(...) [T]hose extensions seem trivial as a contribution of the paper, since they only build on existing techniques. (...)"
>
> The contribution might seem trivial from a technical perspective, however, it was not attempted before to build a faceted navigation application around HDT. HDT was intended to be a format in order to publish and exchange RDF data in a compressed manner. If it is used in this way, it is thus convenient to explore unknown RDF data sets in HDT format without needing to decompress them. Additionally, HDT allows the efficient resolution of triple patterns in-memory. In many cases the size of the HDT file and our indices is much smaller than the original Turtle dump such that it can be efficiently explored also on machines with little RAM capacity.
>
>     "Commentary for Fig 5 and Fig 6 could be improved. For example, why 139\% of the Reactome turtle file is needed? (...)"
>
> It is true, this explanation is important. Due to the limitation of 15 pages, we probably had to prune on some sections a little bit too much. The Reactome data set has a high interconnectivity, i.e., the data set has many triples $(s, p, o)$ where $s$ and $o$ have a respective RDF class assigned. Therefore, the indices are comparatively large on this data set.
>
>     "The actual related work is covered only in a single paragraph, without appropriately explaining why this approach is better."
>
> Indeed due to the space limitations, we described related approaches rather briefly, but note also that this is the first approach to evaluate faceted navigation operations over HDT, so while there are "related application", there is no directly related approach. This extends to the evaluation too, where it is difficult to determine if an approach is "better". Other faceted navigation applications have different exploratory operations, e.g., GraFa uses the available filters but not the reachable classes operation. Also these exploratory operations sometimes only consider result estimations and not exact results like our approach.
>
>     "Also, I do not see actually the benefit of your approach when compared to approaches without the proposed indexes. In some cases what you propose is better but in others your proposal is worst."
>
> We could not identify an experiment where our approach performed considerably worse than the other deployed approaches. But it is true, the benefit of the BT indices in case of the filter queries is questionable. Nevertheless, we think it is important to report these results. Note that in all other experiments, however, our approach performed very well in that we had a maximum execution time of 3 seconds and the other approaches had at least a timeout.
>
>     "Assuming that you selected to present the best score the experiments show that your proposal need to be certainly improved!"
>
> We included the results of the four largest data sets for which our indices could be generated with the available hardware and not the best scores. We further use the principle of stratified randomisation for the queries to provide a fair evaluation and we report also on the results for the filter queries, which were not as good as for the other tested scenarios. It would have certainly possible to "tune" the evaluation results, but we think it is important to give a complete picture, which is also why we included YAGO where the high number classes renders our approach unsuitable.
>
>     "Showing only that your proposal is feasible in most of the cases (you even fail to deliver results for YAGO) is not enough, especially when you don’t have sufficient explanations on why this is happening."
>
> We apologize if the explanation "For the YAGO 2 data set, the CtC and PV indices could not be generated because of insufficient main memory capacity for the high number of distinct RDF classes." was not sufficiently clear. It might have helped to point out that the current generation method in case of the class-to-class indices has a quadratic time- and space- complexity, which is not feasible with the 373,442 RDF classes in YAGO.
> For a smaller number of classes the quadratic complexity is not an issue.
>
>     "I would try to identify where your approach is better, highlighting this aspect right from the introduction. In addition, your paper could be improved by adding a comparison by another faceted search system somehow."
>
> (...)
>
>     "A comparison with another faceted search system would enhance the quality of the paper."
>
> Our approach shows that faceted navigation is feasible directly over HDT files, which is possible due to the indices we propose, and we evaluate the kinds of operations required by typical faceted search applications.  Note that we do not aim for "providing" another faceted search application, but we aim at showing whether and how HDT files can directly be used in faceted exploration operations. We will try to point this out more clearly already in the introduction.

---

### Official Review · AnonReviewer5 · 2021-01-14
**Good idea for an alternative use of HDT for data exploration**

**Confidence:** 5
**Impact:** 3
**Design And Technical Quality:** 3

**Review:**

The paper tackles the issue of efficient faceted navigation on RDF graphs, which results generally costly for large datasets. Authors propose to speed up faceted queries by creating additional partial indexes using the well-known HDT compression (self-indexing) technique. To do so, authors reuse the dictionary and ID-based triples components of HDT, and they add additional ID-based triples indexes (i.e. HDT Bitmap Triples, BTs), two per class. These partial indexes include incoming triples and outgoing triples for a class, respectively. In addition, they include a third class-to-class partial index. Given that the Bitmap triples require a correlative order (which is broken by the partiality), they include a small structure to retrieve the original data. Experiments fill made up scenarios shows that HDT and the partial indexes speed up faceted navigation over HDT alone, being competitive with a commercial solution on some tasks. The size overhead does not double the size of the original HDT.

The paper id generally very well written and tackles an interest and relevant topic. The idea of reusing the HDT BTs for partial indexes is novel and is worth to be pursued. I have however some concerns on the technical design and evaluation:

- Acknowledging that the approach for extending BT is simple, it is also not efficient in space, as triples are, literally repeated. Given that the triples are already existing in HDT, wouldn't it be more efficient to create posting lists for each class? For example, for the outgoing Property-Value index, knowing the list of ID subjects would be enough, as one could jump to the triples in HDT directly. If entering by predicate is needed, then this list can be ordered / split by predicate ID, so one can first order the subjects for predicate 1, then for predicate 2, etc. The approach for incoming property-value could be more tricky, but it might also go along these lines.

- The paper has a big gap IMHO and it is the lack of clarity regarding the practical use of the indexes. When are the new indexes used and when are the traditional ones? I would expect some formalisms at least for triple pattern matching using the new indexes. Otherwise one can only guess when the new ones are used.

- As for the evaluation, I do not totally understand why not all systems are used in all tasks. What makes the system X do not suitable for some of them? Also, I would suggest to use an open source system (e.g. open source virtuoso) rather than a mysterious one to which one cannot compare or redo the experiments.






**Anonymity:**

Yes, I would like my review to remain anonymous.

**Rating:**

-1: Weak Reject

**Reuse And Availability:**

5: Very High

**Strong Points:**

- Reuse of a "bullet proofed" technique
- Simple solution
- Novel benchmark design

**Subreviewer:**

I submitted this review.

**Weak Points:**

- Unclarity on the use of the indexes in practice
- Inefficient design of the partial indexes
- Unclear evaluation

---

> ### Author Rebuttal · Authors · 2021-01-29
>
>     "Acknowledging that the approach for extending BT is simple, it is also not efficient in space, as triples are, literally repeated."
>
> Sure, but that's a well accepted trade-off in database technology in general. For query intensive scenarios (OLTP) the indexes are often larger than the data itself for the sake of performance. In our approach this only occurred for the Reactome dataset.
> One possible optimization is indeed to store all triples exactly once by introducing "representatives", i.e., indices which store the triples for specific RDF class combinations. To fetch all relevant indices (i.e. representatives) for a given RDF class, a mapping from this class to all representatives is required.
>
>     "Given that the triples are already existing in HDT, wouldn't it be more efficient to create posting lists for each class? For example, for the outgoing Property-Value index, knowing the list of ID subjects would be enough, as one could jump to the triples in HDT directly. If entering by predicate is needed, then this list can be ordered / split by predicate ID, so one can first order the subjects for predicate 1, then for predicate 2, etc. The approach for incoming property-value could be more tricky, but it might also go along these lines."
>
> In case of the outgoing PV indices this approach could indeed work well if we only want to iterate over all triples for a given predicate (i.e. $(p, ?s, ?o)$ query). However, in our implementation, we required a PO-S index for the outgoing PV indices in order to fetch all subjects which are annotated by a given property-value pair (i.e. a $(p, o, ?s)$ query).
>
>     "The paper has a big gap IMHO and it is the lack of clarity regarding the practical use of the indexes. When are the new indexes used and when are the traditional ones? I would expect some formalisms at least for triple pattern matching using the new indexes. Otherwise one can only guess when the new ones are used."
>
> Indeed, this should have been described in more detail. For the class queries, we do not require the original HDT BT indices or the FoQ index - we only use the CtC indices. On the other hand, in case of the filter queries, we used a heuristic threshold $t$: Given a set of resources $M$, if $size(M) < t$, we use the HDT FoQ index to test the filter condition by executing a triple pattern query for each resource $s \in M$. Otherwise, we iterate over the respective triples from the PV indices and check the condition.
>
>     "As for the evaluation, I do not totally understand why not all systems are used in all tasks. What makes the system X do not suitable for some of them? Also, I would suggest to use an open source system (e.g. open source virtuoso) rather than a mysterious one to which one cannot compare or redo the experiments."
>
> SystemX (that is RDFox) is freely available with an academic license, but it is required to submit a planned publication 4 weeks prior to the submission deadline to RDFox so that the RDFox team has a chance to point out misconfigurations of the system or analyse the problems. Unfortunately, we could not finish the paper 4 weeks ahead of the deadline and agreed with the PC chairs that we initially submit a version where the system's name is anonymised until we get the official RDFox approval, which we have by now. We consider RDFox an interesting choice as it also provides the underlying functionalities needed for faceted search.
> RDFox and Jena were not included in the available filters and the reachable classes experiment since these operations are executed relative to a given center. We wanted to compare only the required time for the specific operation and therefore these measurements should not be distorted by the initial class query to get the centre. But of course, this is a valid point and it is indeed interesting how other approaches perform on these operations. Our main goal was, however, to examine if our given BT indices approach outperforms the plain HDT approach which solely uses the HDT file and the FoQ index.

---

> > ### Comment · AnonReviewer5 · 2021-01-29
> > **Thanks for the rebuttal**
> >
> > I would like to acknowledge and thanks the comments of the authors. Nonetheless I will keep my previous scores given two main facts also acknowledged by authors in their rebuttal: the lack of formalisms or descriptions of the actual use of the indexes in the queries, and the important gaps in the evaluation

---

### Official Review · AnonReviewer2 · 2021-01-14
**Solid paper on an important topic, though with modest novelty and some key limitations**

**Rating:** 1
**Confidence:** 4
**Impact:** 3
**Design And Technical Quality:** 4

**Review:**

# Summary

The paper describes in-memory indexing approaches to support efficient faceted browsing over RDF graphs. The proposed approach is based on HDT, upon which specialised indexes are built, including a sub-graph index for the current focus of the browsing, indexes on incoming and outgoing edges for particular classes, and a "class-to-class" index that covers triples where the subject and object are instances of a given pair of classes. The proposed indexing scheme is evaluated over a variety of RDF graphs (up to 150 million triples), being compared with Jena ARQ using a HDT backend, a commercial in-memory RDF store, and various configurations of the indexes proposed. The results show that for more complex queries, particularly those that take as input an existing centre, the various indexing schemes show more stable performance that the baseline methods.

# Strengths (summary)

1. Addresses an important challenge (usability / efficient faceted browsing).

2. The paper is well written.

3. The work itself is well-executed with solid experiments, etc.

# Weaknesses (summary)

1. The delta of the paper is relatively minor versus existing methods.

2. The system does not scale particularly well (150 million triples).

3. Some design choices (e.g., building upon HDT), though perhaps reasonable, are not well-justified.

# Verdict

Overall, though not overly ambitious in scope, I think this is a solid, well-written, well-executed paper on an important topic. I lean towards accepting. I think the paper could benefit from being more clear about the reasons for the scale limitations, and also being more explicit about some of the design choices made.


**Anonymity:**

No, I would like my review to be deanonymized.

**Reuse And Availability:**

5: Very High

**Strong Points:**

1. Usability is an important challenge for the SW community. Efficient faceted browsers offer an excellent way for users to explore RDF graphs. Hence there is a lot of value in the line of research that the authors embark on here.

2. The paper is well written, with clear formal definitions, complemented with good examples to help understand the technical aspects.

3. The proposed techniques, though perhaps not ground-breaking, are quite solid. The experiments consider a variety of different dataset and query types. There are one or two minor technical details to revise (see minor comments below), but overall these details make sense.

**Subreviewer:**

I submitted this review.

**Weak Points:**

1. The novel technical contributions are some fairly minor extensions over HDT in order to support indexing sub-graphs that reference (use the same IDs as) the full graph, as well as a couple of custom filtered indexes. The technical description of the delta is described in around 1.5 pages of the paper. These extensions do provide performance benefits per the experimental results, but I think it is fair to say that the delta over existing methods is relatively minor overall.

2. The paper mentions that there are scalability issues with the proposed approach, and experiments extend to graphs with a scale of around 150 million triples, which is quite modest by today's standards. Many of the most prominent RDF graphs (DBpedia, Wikidata, LinkedGeoData) greatly exceed this scale. There is not much discussion of this limitation. Is it purely an issue of RAM? With more RAM would larger graphs be supported? Also there are limitations regarding the number of classes, as one of the indexes creates a structure that is quadratic in size with respect to the number of classes (which becomes an issue for indexing YAGO). Thus the proposed indexes would not (yet) be applicable in many (important) scenarios.

3. The proposed indexes are built upon HDT, but this design choice is not entirely justified. Could other compact data structures have enabled better performance or smaller indexes? Were other alternatives considered?

---

# Minor comments
- "reserved as [a] wildcard character"
- "each partial object sequence[]"
- "the length of $S^i_p$ and $S^j_o$" It is not immediately clear if this in terms of symbol length, character length, bit length, etc. (though it becomes clear later).
- From the discussion of the example of Figure 3, without the dictionary, is it not clear to me why 1 is a shared resource, but 2 and 3 are pure objects.
- Section 3: I guess that the idea of indexing a subset of the data is to index the current centres. It would be good to make this explicit to put the proposal into context.
- Figure 4: Should $\hat{S}_o$ contain 1? Isn't this an object of node 2?
- Definition 9: "index over[ ]{...}"
- "We generate CtC indices for each pair of classes C" This being quadratic in the number of classes would be problematic for datasets such as Wikidata with thousands of classes, though I guess if it were restricted to C and D such that there exists $(s,p,o) \in G$ with $s$ of type $C$ and $o$ of type $D$ it should be fine in practice though still potentially quadratic on the number of classes. (Indeed these problems arise with YAGO later in the experiments.)
- Where does owl:NamedIndividual appear from if RDFS reasoning is applied? Wouldn't it be rdfs:Resource?
- In Figure 6, I am not convinced about the stacked representation as the .ttl file does not add to the whole. It would seem more reasonable to have a .ttl bar and the rest in a stack alongside.
- "A fair comparison with the other approaches, which execute SPARQL queries, would be difficult in this scenario" What about using VALUES to pass the current centre in the query? (Admittedly it might generate large SPARQL query strings that systems are not good at handling.)

---

> ### Author Rebuttal · Authors · 2021-01-29
>
>     "Some design choices (e.g., building upon HDT), though perhaps reasonable, are not well-justified."
>
> It is true, we could point that out more clearly. HDT was intended to be a format in order to publish and exchange RDF data in a compressed manner. If it is used in this way, it is thus convenient to explore unknown RDF data sets in HDT format without needing to decompress them. Additionally, HDT allows the efficient in-memory resolution of triple patterns. On machines with little RAM capacity, the size of the HDT file and the indices must be sufficiently small for an efficient RDF data set exploration.
>
>     "These extensions do provide performance benefits per the experimental results, but I think it is fair to say that the delta over existing methods is relatively minor overall."
>
> Compared to existing methods, we indeed lack any query planning and our method, the number of RDF classes is a limiting factor for the proposed index strategy. Note also that a direct comparison with existing exploration methods is difficult since the concrete exploratory operations vary significantly between different faceted navigation applications, as e.g.:
> - available filters (e.g. GraFa) vs. reachable classes (e.g. SemSpect)
> - exact number of results (e.g. GraFa, eLinda, our approach) vs. partial results (e.g. Sparklis)
>
>
> Note that, in particular, the use of approximate results allows for handling much larger data sets.
> Furthermore, many approaches from other faceted navigation applications are evaluated solely over a single data set (e.g. GraFa) and, in comparison to our benchmark, we could not find an approach were an attempt was made to extract queries which are representative for a given RDF data set.
>
>     "The paper mentions that there are scalability issues[...]. Many of the most prominent RDF graphs (DBpedia, Wikidata, LinkedGeoData) greatly exceed this scale. There is not much discussion of this limitation. Is it purely an issue of RAM? With more RAM would larger graphs be supported? Also there are limitations regarding the number of classes, as one of the indexes creates a structure that is quadratic in size with respect to the number of classes (which becomes an issue for indexing YAGO)."
>
> Indeed the scale limitations could have been explained more clearly. The YAGO2 data set is an exception with its 373,442 RDF classes. Concerning this data set, a generation method with quadratic time- (and space-) complexity is not suited. In all other cases, more RAM definitely allows the generation of the BT indices concerning data sets with more triples.
>
>     "The proposed indexes are built upon HDT, but this design choice is not entirely justified. Could other compact data structures have enabled better performance or smaller indexes? Were other alternatives considered?"
>
> We only experimented with HDT and extended it by the indices in the BT index format. Initially, we considered implementing an index similar to HDT-FoQ (i.e., storing references to the triples of the given HDT file), but the current BT index implementation has the benefit that triple pattern queries can be resolved more easily and also more options are available concerning the triple component order (e.g., PS-O, PO-S etc.).

---

> > ### Comment · AnonReviewer2 · 2021-01-31
> > **Correcting error in my rating and responding to rebuttal**
> >
> > Firstly, my sincere apologies for the incorrect rating. I had intended to assign this paper a weak accept as indicated in the Verdict where I said "I lean towards accepting". Somehow I uploaded the original review selecting the incorrect rating.
> >
> > I thank the authors for the clarifications in the rebuttal. Overall the response confirms and justifies some current limitations of the approach. I think adding the indicated clarifications to the paper would be useful. Considering the rebuttal and the other reviews, while the approach has some limitations and potentially could be improved in future, importantly I think that the paper and the authors have been quite transparent about these limitations, and thus I think this is a useful work that could lead to future improvements. Overall I think that "Weak accept" remains a fair recommendation.

---

### Decision · Program_Chairs · 2021-02-23

**Decision:**

Accept

**Comment:**

There were essentially two fronts on this heavily debated paper in the reviews, which both have their pros and cons:

- on the one hand, the negative reviews criticized mainly the indexing method used a marginal contribution/improvement with regards to indexing methods as such above the state of the art, and viewed the "faceted browsing" more like a sales argument for an otherwise unspectacular indexing approach.
vs.
- on the other hand, the positive reviewers emphasized that this is an interesting paper for using its method for faceted browsing of KGs as such, a task that the authors credibly emphasize that current standard triple stores struggle with and they can scale to reasonable sizes of graphs up to 150M triples, where the suggested approach of leveraging HDT plus BT indexes could serve as a basis with potential further future improvements and where the the paper points to a new, so far not really much explored *application direction* of tailored indexing methods in the otherwise not so much researched area of graph browsing and visualization tooling.

After reading all the verdicts and looking at the paper as a whole again, we decided to accept this paper as an indeed controversial paper, that hopefully sparks new ideas.